# FEDLORA: WHEN PERSONALIZED FEDERATED LEARNING MEETS LOW-RANK ADAPTATION

## ABSTRACT

In this research paper, we introduce a novel approach to Personalized Federated Learning (PFL), which we call FedLoRA. This approach is inspired by recent advancements in fine-tuning Large Language Models (LLMs), particularly the Low-Rank Adaptation (LoRA) technique. The remarkable success of LoRA demonstrates that general linguistic knowledge is preserved in a pre-trained full-rank model, while domain-specific knowledge can be effectively retained within a low-rank parameter matrix. Building upon this insight, we present FedLoRA in the context of PFL, aiming to maintain shared general knowledge among all clients in a common full-rank matrix, while capturing client-specific knowledge within a personalized low-rank matrix. However, the integration of LoRA into PFL presents its own set of challenges. Unlike LoRA, which starts with pre-trained general knowledge, FedLoRA's full-rank matrix needs training from scratch. This phase can be notably influenced by data heterogeneity, potentially hindering its effective extraction of general knowledge. To address this challenge, we propose a new training strategy to mitigate the effects of data heterogeneity on the shared full-rank matrix. Our experimental results, obtained across multiple datasets exhibiting varying degrees of data heterogeneity, demonstrate that FedLoRA outperforms current state-of-the-art methods significantly.

## 1 INTRODUCTION

Federated learning (FL) McMahan et al. (2016) allows clients to collaboratively train a global model without directly sharing their raw data. A central challenge in FL is data heterogeneity, where the data distributions across diverse clients are not independently and identically distributed (non-IID). Such disparities in data distributions hamper the training of the global model, leading to a decrease in the performance of FL Zhang et al. (2021a); Gong et al. (2021); Li et al. (2021a).

To confront this challenge, the concept of Personalized Federated Learning (PFL) has been introduced. Within PFL studies, it is widely accepted that the knowledge learned by each client can be divided into general knowledge and client-specific knowledge. This understanding prompts mainstream PFL research to split a model into two parts: a global part with shared parameters across all clients to preserve general knowledge and a personalized part with unique parameters for each client to retain client-specific knowledge. The task of decomposing a model into its shared and personalized components is garnering increasing attention. Numerous related studies have surfaced, with several focusing on the personalization of parameters in specific layers. For instance, FedPer by Arivazhagan et al. (2019) focuses on personalizing the classifier, whereas FedBN by Li et al. (2021c) targets the personalization of Batch Normalization layers. Another strand of PFL research introduces extra personalized layers to the original model Pillutla et al. (2022); Zheng et al. (2022).

Concurrently, the fine-tuning of large language models (LLMs) has emerged as another field garnering attention. Fine-tuning aims to enhance a pre-trained language model (which contains broad general linguistic knowledge), by incorporating domain-specific knowledge, thereby making it more suitable for specific downstream tasks. We note that LLM fine-tuning aligns closely with PFL in several dimensions. Firstly, examining their intrinsic purposes, the role of the pre-trained model in LLM aligns with the shared component in PFL, as both are concentrated on the preservation of general knowledge. In parallel, the fine-tuning phase of LLM corresponds with the personalized component of PFL, with a mutual objective of preserving domain-specific knowledge. Beyond their

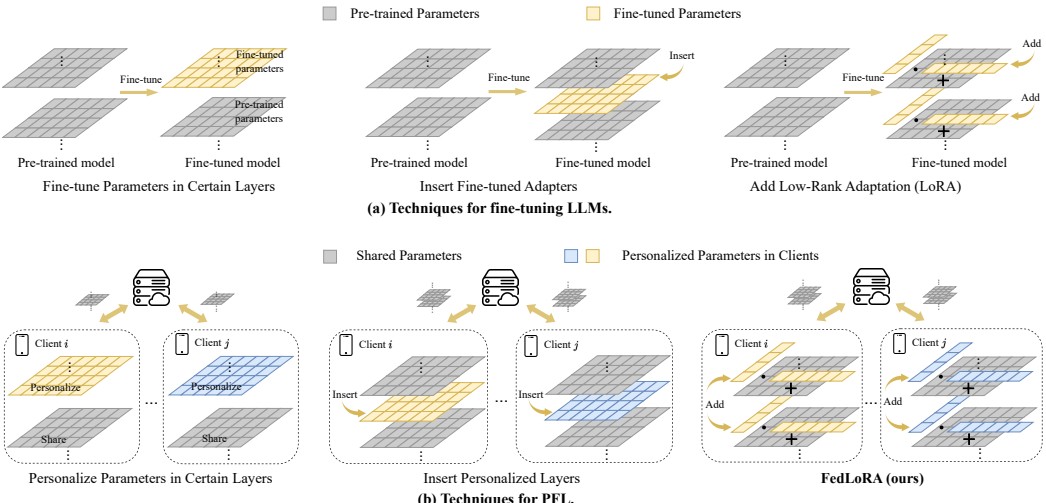

Figure 1: The development of related techniques in LLM fine-tuning and PFL.

objectives, the methodologies employed in LLM fine-tuning and PFL also share similarities, as depicted in Fig. 1. A summary of various LLM fine-tuning approaches is showcased in Fig. 1(a). For example, one prevalent method emphasizes fine-tuning parameters in specific layers Kenton & Toutanova (2019); Zaken et al. (2021). Other techniques introduce additional trainable parameters, known as "adapters", into LLMs; notable examples include AdapterDrop by Rücklé et al. (2020) and AdapterFusion by Pfeiffer et al. (2020). From the above discussion, it becomes clear that, despite the differences in the training processes between fine-tuning LLMs and PFL, the approaches to parameter decomposition exhibit remarkable parallels.

Besides the approaches mentioned above, the state-of-the-art LLM fine-tuning method is Low-Rank Adaptation (LoRA) Hu et al. (2022). As shown in the right of Fig. 1(a), LoRA acquires domain-specific knowledge through a low-rank parameter matrix, while general knowledge is preserved in a full-rank parameter matrix. Inspired by this insight, we propose FedLoRA as an analogous approach in FL. FedLoRA decomposes each personalized model layer into a shared full-rank parameter matrix and a personalized low-rank parameter matrix. The former retains general knowledge ubiquitous across clients, whereas the latter concentrates on client-specific knowledge.

However, integrating LoRA into PFL introduces distinct challenges. Unlike LoRA, initiated with a pre-trained general model, FedLoRA needs to train the full-rank parameter matrix from scratch, which can be substantially influenced by non-IID data. To address this issue, we investigate the order of training between the full-rank and low-rank parameter matrices during local updating. Specifically, this involves initially training the low-rank part to diminish the influence of non-IID data, followed by the training of the full-rank part. Our findings indicate that unlike previous methods that concurrently train both parameter matrices, adopting an alternating approach is more beneficial.

Our primary contribution in this paper can be summarized as follows:

- Inspired by LoRA in the area of fine-tuning LLMs, we introduce a new method of decomposing shared and personalized parameters in PFL, namely FedLoRA. Specifically, we decompose each layer of the personalized model into a shared full-rank part to preserve general knowledge, and a personalized low-rank part to preserve client-specific knowledge.

- We introduce an innovative training strategy designed to optimize FedLoRA, effectively mitigating the implications of non-IID data and significantly boosting performance.

- We evaluate FedLoRA across multiple datasets and under varied non-IID conditions. Our findings underscore the efficacy of the FedLoRA method we propose.

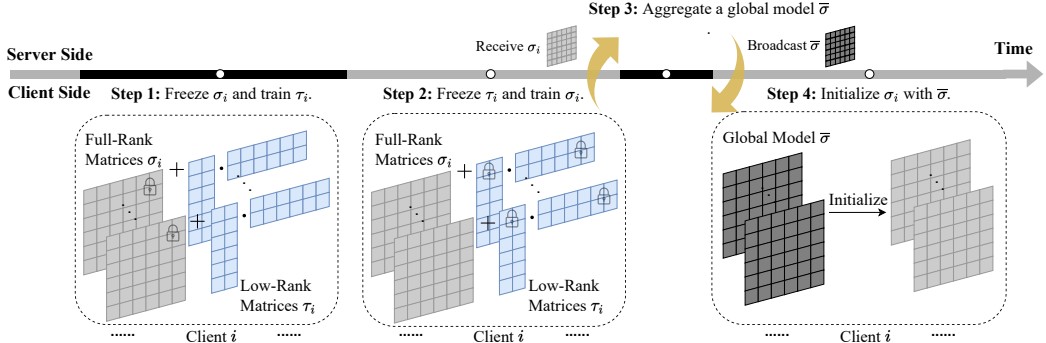

Figure 2: The overview of one client in FedLoRA in one communication round.

## 2 RELATED WORK

PFL has emerged as a prevalent research direction to handle the non-IID problem in FL. Current PFL methods can be mainly divided into meta-learning-based methods Fallah et al. (2020); Acar et al. (2021), fine-tuning-based methods Jin et al. (2022); Chen et al. (2023), model-regularization-based methods T Dinh et al. (2020); Li et al. (2021b), personalized-aggregation-based methods Huang et al. (2021); Zhang et al. (2021b), and parameter-decomposition-based methods. Among them, parameter-decomposition-based methods, which decompose models into a shared part and a personalized part, are most relevant to ours.

**Personalize certain parameters within the original model.** The core idea of this kind of method is to share part of the original model's parameters while personalizing the other part. Representative works include selecting specific layers for personalization, such as FedPer and FedRep Arivazhagan et al. (2019); Collins et al. (2021) proposing to personalize classifiers, and FedBN Li et al. (2021c) suggesting to personalize the Batch Normalization (BN) layers. Other works employ Deep Reinforcement Learning (DRL) or hypernetworks technologies to automate the selection of specific layers for personalization Sun et al. (2021); Ma et al. (2022). Still, some other research no longer selects personalized parameters based on layers but on each individual parameter, making more fine-grained choices to personalize parameters sensitive to non-IID data Wu et al. (2023).

**Add extra personalization layers to the original model.** In recent years, some studies propose another kind of personalized parameter partitioning method. Unlike the previous method, the core idea of this method is to add additional personalized layers to the original model. For example, ChannelFed Zheng et al. (2022) introduces a personalized attention layer to redistribute weights for different channels in a personalized manner. Pillutla et al. (2022) proposes to add a bottleneck module for personalization after each feedforward layer.

Different from the prior research, our paper introduces a new perspective on parameter decomposition. We embed general knowledge into a full-rank matrix and client-specific knowledge into a low-rank matrix to achieve better knowledge sharing across clients and personalization.

## 3 METHOD

### 3.1 OVERVIEW OF FEDLORA

We first give an overview of FedLoRA. As illustrated in Fig. 2, each layer of client $i$'s personalized model is decomposed into the sum of a full-rank matrix and a low-rank matrix. The training process in each communication round can be summarized as follows: 1) each client $i$ freezes its full-rank matrices $\boldsymbol{\sigma}_i$ and updates the low-rank matrices $\boldsymbol{\tau}_i$. 2) Then, each client $i$ turns to update $\boldsymbol{\sigma}_i$ and freeze $\boldsymbol{\tau}_i$. After local updating, all clients upload the full-rank part to the server while keeping the low-rank part private. 3) The server receives clients' parameters and aggregates them to generate a global model $\overline{\boldsymbol{\sigma}}$. After doing this, the server sends $\overline{\boldsymbol{\sigma}}$ back to all clients. 4) Each client receives the global model and uses it to initialize the full-rank matrices.

## 3.2 PROBLEM DEFINITION OF PFL

PFL, in contrast to traditional FL algorithms that train a universal model for all clients, strives to develop a personalized model for each client $i$, denoted as $w_i$, specializing in capturing the unique characteristics of its local data distribution $D_i$. In recent PFL research, there is a consensus that the knowledge acquired by individual clients comprises both general knowledge and client-specific knowledge. In non-IID scenarios, since different clients have distinct data distributions (i.e., $D_i \neq D_j, i \neq j$), it is difficult to extract general knowledge and thus brings challenges to client collaboration. To address this problem, PFL decouples $w_i$ into a shared part $\sigma$ and a personalized part $\tau_i$ to learn general knowledge and client-specific knowledge respectively. Formally, the training objective can be formulated as

$$\min_{\sigma, \tau_1, \tau_2, \ldots \tau_N} \sum_{i=1}^{N} L_i(\sigma; \tau_i; D_i), \tag{1}$$

where $L_i(\sigma; \tau_i; D_i)$ denotes the loss function of client $i$ and $N$ is the total number of clients. To optimize the target function in equation 1, recent studies have put forth various PFL methods to partition $\tau_i$ and $\sigma$. While these endeavors have shown promise, the question of how to further refine the decomposition of these two parameter components still presents an unresolved challenge.

## 3.3 LOW-RANK PARAMETER DECOMPOSITION

To develop an efficient parameter decomposition method, we draw inspiration from recent advancements in fine-tuning LLM, specifically a technique known as LoRA. LoRA suggests a novel approach where domain-specific knowledge is embedded into a low-rank parameter matrix, while general linguistic knowledge remains integrated into a pre-trained model with a full-rank parameter matrix. Building upon this concept, we propose a similar approach within the context of PFL. In PFL, we observe that shared parameters responsible for extracting general knowledge benefit from a high model capacity. In contrast, personalized parameters are tasked with learning knowledge that complements the general understanding for specific local tasks (i.e., client-specific knowledge), therefore, it is sufficient to use a low-rank matrix to represent these personalized parameters. Formally, in FedLoRA, we assume that each personalized model has a set of weights $\boldsymbol{\theta_i} = \{\theta_i^k\}_{k=1}^{L}$, where $\theta_i^k$ is the weights for the $k$-th layer and $L$ is the total layer number. Each $\theta_i^k$ is decomposed as

$$\theta_i^k = \sigma_i^k + \tau_i^k, k \in [1, L], \tag{2}$$

where $\sigma_i^k, k \in [1, L]$ is a full-rank parameter matrix that is shared across all clients, and $\tau_i^k, k \in [1, L]$ is a personalized low-rank parameter matrix. In the following, we employ the notation $\boldsymbol{\theta_i}, \boldsymbol{\sigma_i}$ and $\boldsymbol{\tau_i}$ to denote the complete model parameter set, the full-rank parameter set, and the low-rank parameter set specific to client $i$, respectively. Additionally, we use $\theta_i^k, \sigma_i^k$, and $\tau_i^k$ to represent the parameter matrices for layer $k$ within client $i$.

Next, we present the methods for imposing low-rank constraints on $\boldsymbol{\tau_i}$.

**Low-rank Decomposition of Fully-Connected Layers:** For fully-connected layers, we follow the decomposition method outlined in LoRA. The dimension of $\tau_i^k$ is $I \times O$, where $I$ and $O$ represent the input and output dimensions. We constrain $\tau_i^k$ through a low-rank decomposition as follows:

$$\tau_i^k = B_i^k A_i^k, \text{where } B_i^k \in \mathbb{R}^{I \times (R_l \cdot \min(I, O))} \text{ and } A_i^k \in \mathbb{R}^{(R_l \cdot \min(I, O)) \times O}. \tag{3}$$

The $R_l$ serves as a hyper-parameter designed to regulate the rank of $\tau_i^k$ within fully-connected layers. Its value falls within the range of $0 < R_l \leq 1$.

**Low-rank Decomposition of Convolutional Layers:** In contrast to fully-connected layers, convolutional layers involve multiple kernels, resulting in $\tau_i^k \in I \times O \times K \times K$ dimensions. However, we can still apply a low-rank decomposition to constrain $\tau_i^k$ as follows:

$$\tau_i^{k*} = B_i^k A_i^k \in \mathbb{R}^{(I \cdot K) \times (O \cdot K)}, \tag{4}$$

$$\text{where } B_i^k \in \mathbb{R}^{(I \cdot K) \times (R_c \cdot \min(I, O) \cdot K)} \text{ and } A_i^k \in \mathbb{R}^{(R_c \cdot \min(I, O) \cdot K) \times (O \cdot K)},$$

$$\tau_i^k = \text{Reshape}(\tau_i^{k*}) \in \mathbb{R}^{I \times O \times K \times K}. \tag{5}$$

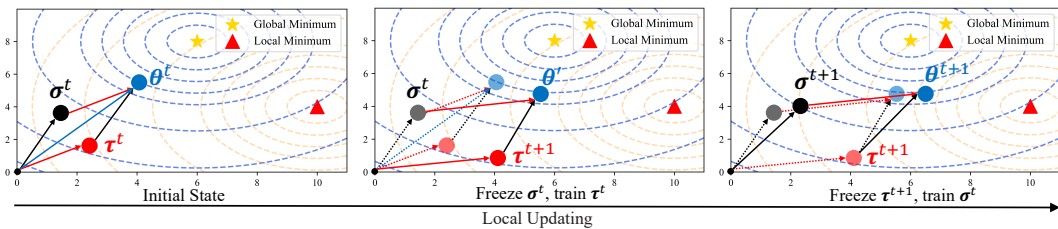

Figure 3: A toy example to illustrate the alternating training in FedLoRA.

The $R_c$ is a hyper-parameter used to control the rank of $\tau_i^k$ within convolutional layers. Its value is within the range of $0 < R_c \leq 1$.

During training, both $B$ and $A$ serve as trainable parameter matrices. We initialize $A$ with random Gaussian values and $B$ with zeros, which means $\tau_i^k$ starts as zero at the beginning of training.

The hyper-parameters $R_l$ and $R_c$ play crucial roles in controlling the rank of parameters within fully connected and convolutional layers, respectively. As the rank increases, the learning capacity of personalized parameters within the model gradually improves. However, if the rank is set too low, $\tau_i$ may struggle to effectively capture client-specific knowledge, making $\sigma_i$ highly susceptible to non-IID data distributions. This, in turn, negatively impacts collaboration among clients. In contrast, if the rank is too large, $\tau_i$ may start to absorb some of the general knowledge that should be learned by $\sigma_i$, diminishing the level of collaboration among clients. For simplicity, in the FedLoRA approach, we apply the same $R_c$ to all convolutional layers and the same $R_l$ to all fully-connected layers. This simplification streamlines the model architecture and hyper-parameter tuning process.

### 3.4 COORDINATE TRAINING BETWEEN $\sigma$ AND $\tau$

To better extract general knowledge, in contrast to the common practice where personalized and shared parameters are trained simultaneously, we find that a more effective strategy is to initially train the low-rank parameters. This alternating approach helps mitigate the impact of non-IID data before proceeding to train the full-rank parameters. Formally, in each communication round $t \in [1, T]$, we first optimize the low-rank parameters $\tau_i$ for $E_{\text{lora}}$ epochs by

$$\tau_i^{t+1} = \arg\min_{\tau_i} L_i(\tau_i^t, \sigma_i^t, D_i). \tag{6}$$

Then optimize the full-rank parameters $\sigma_i$ for $E_{\text{global}}$ epochs by

$$\sigma_i^{t+1} = \arg\min_{\sigma_i} L_i(\tau_i^{t+1}, \sigma_i^t, D_i). \tag{7}$$

We set $E_{\text{lora}} + E_{\text{global}} = E$, where $E$ is the total number of local update epochs in one round. These hyper-parameters play an important role in balancing the learning dynamics between two key components, $\sigma_i$ and $\tau_i$. When $E_{\text{lora}}$ is set higher, it results in $\sigma_i$ learning less knowledge. Consequently, the degree of knowledge sharing among clients diminishes. In contrast, if $E_{\text{lora}}$ is set too low, $\sigma_i$ ends up acquiring a significant amount of client-specific knowledge. This scenario increases the risk of clients sharing knowledge that is more susceptible to non-IID data. In special cases, when $E_{\text{lora}} = 0$, the FedLoRA framework degenerates into FedAvg. Similarly, when $E_{\text{global}} = 0$, FedLoRA transforms into local training with low-rank parameters, without any collaborative efforts among clients.

After local updating, each client $i$ uploads $\sigma_i^{t+1}$ to the server while keeping the $\tau_i^{t+1}$ private. The server computes a global model $\overline{\sigma}^{t+1}$ by aggregating all clients' $\sigma_i^{t+1}$ through

$$\overline{\sigma}^{t+1} = \frac{1}{N} \sum_{i=1}^{N} \sigma_i^{t+1}, \tag{8}$$

and sends it back to clients. The detailed training process is summarized in the Algorithm 1.

To explain our intuition for proposing alternating training, we employ a toy example to illustrate the local update phase of each client's personalized model within the parameter space. As shown in

---

**Algorithm 1** FedLoRA

---

**Input:** Each client's initial personalized parameter matrices $\boldsymbol{\tau}_i^1$; The global shared parameter matrices $\overline{\boldsymbol{\sigma}}^1$; Number of clients $N$; Total communication round $T$; Global matrices update epoch number $E_{\text{global}}$; LoRA matrices update epoch number $E_{\text{lora}}$ ;

**Output:** Personalized model parameter matrices $\boldsymbol{\theta}_i^T$ for each client.

**for** $t = 1$ to $T$ **do**
    **Client-side:**
    **for** $i = 1$ to $N$ **in parallel do**
        Initializing $\boldsymbol{\sigma}_i^t$ with $\overline{\boldsymbol{\sigma}}^t$.
        Updating $\boldsymbol{\tau}_i^t$ by equation 6 for $E_{\text{lora}}$ epochs to obtain $\boldsymbol{\tau}_i^{t+1}$.
        Updating $\boldsymbol{\sigma}_i^t$ by equation 7 for $E_{\text{global}}$ epochs to obtain $\boldsymbol{\sigma}_i^{t+1}$.
        Sending $\boldsymbol{\sigma}_i^{t+1}$ to the server.
    **end for**
    **Server-side:**
    Aggregating a global model $\overline{\boldsymbol{\sigma}}^{t+1}$ by equation 8.
    Sending $\overline{\boldsymbol{\sigma}}^{t+1}$ to each client $i$.
**end for**

---

Fig. 3, the yellow $\star$ and red $\triangle$ denote the optimum points of the global model on all clients' data (global loss minimum point) and the personalized model on the client's data (local loss minimum point), respectively. Under the influence of non-IID, there is a big difference between global knowledge and local knowledge of clients. This makes the local minimum point far away from the global minimum point. The client's personalized model $\boldsymbol{\theta}$ is decomposed into the sum of a shared part $\boldsymbol{\sigma}$ and a personalized part $\boldsymbol{\tau}$. Since we first train the personalized part, the client-specific knowledge is mostly learned by $\boldsymbol{\tau}$ and the shift of $\boldsymbol{\theta}$ to the local minimum point is mainly done by $\boldsymbol{\tau}$. Therefore, when training $\boldsymbol{\sigma}$, it moves less towards the local minimum point (i.e., less affected by non-IID data), so it can better extract general knowledge.

## 4 EXPERIMENTS

### 4.1 EXPERIMENT SETUP

**Dataset.** Our experiments are conducted on three datasets: CIFAR-10 Krizhevsky et al. (2010), CIFAR-100 Krizhevsky et al. (2009), and Tiny ImageNet Le & Yang (2015). To evaluate the effectiveness of our approach in various scenarios, we adopt the Dirichlet non-IID setting, a commonly used framework in current FL research Hsu et al. (2019); Lin et al. (2020); Wu et al. (2022). In this setup, each client's data is generated from a Dirichlet distribution represented as $Dir(\alpha)$. As the value of $\alpha$ increases, the level of class imbalance in each client's dataset gradually decreases. Consequently, the Dirichlet non-IID setting allows us to test the performance of our methods across a wide range of diverse non-IID scenarios. For a more intuitive understanding of this concept, we offer a visualization of the data partitioning in Appendix A.

**Baseline methods.** To verify the efficacy of FedLoRA, we compare it with eight state-of-the-art (SOTA) methods: FedAMP Huang et al. (2021), FedRep Collins et al. (2021), FedBN Li et al. (2021c), FedPer Arivazhagan et al. (2019), FedRoD Chen & Chao (2022), pFedSD Jin et al. (2022), pFedGate Chen et al. (2023), and FedCAC Wu et al. (2023). Among these methods, FedAMP forces clients with similar data distributions to learn from each other. FedBN, FedPer, FedRep, FedRoD, and FedCAC are parameter-decomposition-based methods that personalize parts of the parameters. pFedSD and pFedGate are fine-tuning-based methods, whose goal is to adapt the global model to the client's local data. These methods cover the latest advancements in various directions of PFL.

**Selection for hyper-parameters.** We utilize the hyper-parameters specified in the respective papers for each SOTA method. For the FL general hyper-parameters, we set the client number $N = 40$, the local update epochs $E = 5$, the batch size $B = 100$, and the total communication round $T = 300$. Each client is assigned 500 training samples and 100 test samples with the same data distribution. In each experiment, we select the best mean accuracy across all clients as the performance metric. Each experiment is repeated using three seeds, and the mean and standard deviation are reported. We adopt the ResNet He et al. (2016) network structure. Specifically, we utilize ResNet-8 for CIFAR-10

Table 1: Comparison results under Dirichlet non-IID on CIFAR-10, CIFAR-100, and Tiny Imagenet.

| Method | $\alpha$ | CIFAR-10 | | | CIFAR-100 | | | Tiny Imagenet | | |
|---|---|---|---|---|---|---|---|---|---|---|
| | | 0.1 | 0.5 | 1.0 | 0.1 | 0.5 | 1.0 | 0.1 | 0.5 | 1.0 |
| FedAvg | | 60.39 | 60.41 | 60.91 | 34.91 | 32.78 | 33.94 | 21.26 | 20.32 | 17.20 |
| | | ±1.46 | ±1.36 | ±0.72 | ±0.86 | ±0.23 | ±0.39 | ±1.28 | ±0.91 | ±0.54 |
| Local | | 81.91 | 60.15 | 52.24 | 47.61 | 22.65 | 18.76 | 24.07 | 8.75 | 6.87 |
| | | ±3.09 | ±0.86 | ±0.41 | ±0.96 | ±0.51 | ±0.63 | ±0.62 | ±0.30 | ±0.28 |
| FedAMP | | 84.99 | 68.26 | 64.87 | 46.68 | 24.74 | 18.22 | 27.85 | 10.70 | 7.13 |
| *AAAI 2021* | | ±1.82 | ±0.79 | ±0.95 | ±1.06 | ±0.58 | ±0.41 | ±0.71 | ±0.32 | ±0.21 |
| FedRep | | 84.59 | 67.69 | 60.52 | 51.25 | 26.97 | 20.63 | 30.83 | 12.14 | 8.37 |
| *ICML 2021* | | ±1.58 | ±0.86 | ±0.72 | ±1.37 | ±0.33 | ±0.42 | ±1.05 | ±0.28 | ±0.25 |
| FedBN | | 83.55 | 66.79 | 62.20 | 54.35 | 36.94 | 33.67 | 33.34 | 19.61 | 16.57 |
| *ICLR 2021* | | ±2.32 | ±1.08 | ±0.67 | ±0.63 | ±0.94 | ±0.12 | ±0.71 | ±0.35 | ±0.44 |
| FedPer | | 84.43 | 68.80 | 64.92 | 51.38 | 28.25 | 21.53 | 32.33 | 12.69 | 8.67 |
| | | ±0.47 | ±0.49 | ±0.66 | ±0.94 | ±1.03 | ±0.50 | ±0.31 | ±0.42 | ±0.40 |
| FedRoD | | 86.23 | 72.34 | 68.45 | 60.19 | 38.54 | 33.67 | **44.25** | 27.02 | 22.07 |
| *ICLR 2022* | | ±2.12 | ±1.77 | ±1.94 | ±0.64 | ±1.12 | ±0.48 | **±0.40** | ±0.64 | ±0.79 |
| pFedSD | | 86.34 | 71.97 | 67.21 | 54.14 | 41.06 | 38.27 | 39.31 | 19.25 | 15.91 |
| *TPDS 2023* | | ±2.61 | ±2.07 | ±1.89 | ±0.77 | ±0.83 | ±0.20 | ±0.19 | ±1.80 | ±0.33 |
| pFedGate | | **87.25** | 71.98 | 67.85 | 48.54 | 27.47 | 22.98 | 37.59 | 24.09 | 19.69 |
| *ICML 2023* | | **±1.91** | ±1.61 | ±0.87 | ±0.39 | ±0.79 | ±0.03 | ±0.39 | ±0.67 | ±0.14 |
| FedCAC | | 86.82 | 69.83 | 65.39 | 57.22 | 38.64 | 32.59 | 40.19 | 23.70 | 18.58 |
| *ICCV 2023* | | ±1.18 | ±0.46 | ±0.51 | ±1.52 | ±0.63 | ±0.32 | ±1.20 | ±0.28 | ±0.62 |
| FedLoRA | | 85.47 | **72.78** | **69.09** | **63.65** | **45.96** | **42.98** | 44.22 | **28.25** | **25.55** |
| | | ±2.06 | ±1.23 | ±1.14 | ±0.53 | ±1.19 | ±0.64 | ±0.55 | ±1.24 | ±0.13 |

and ResNet-10 for CIFAR-100 and Tiny ImageNet. In FedLoRA, we adopt the SGD optimizer with a learning rate of 0.1.

## 4.2 COMPARISON WITH SOTA METHODS

In this section, we compare our FedLoRA with several SOTA methods. To ensure a comprehensive evaluation, we consider three different non-IID degrees (i.e., $\alpha \in \{0.1, 0.5, 1.0\}$) on CIFAR-10, CIFAR-100, and Tiny Imagenet.

The results in Table 1 demonstrate that the performance of FedAMP is comparable to other SOTA methods on the CIFAR-10 dataset, but experiences a notable decline on CIFAR-100 and Tiny Imagenet. This is primarily because of its limited capacity to leverage collaboration among clients with diverse data distributions. In contrast, mainstream model decomposition methods such as FedRep, FedBN, FedPer, FedRoD, and FedCAC enhance collaboration among clients by personalizing parameters sensitive to non-IID data while sharing others. Among these methods, FedRoD distinguishes itself by introducing a balanced global classifier to facilitate comprehensive knowledge exchange, underscoring the potential for improvements in client collaboration within current model decomposition strategies. On the other hand, fine-tuning-based approaches like pFedSD and pFedGate enable all clients to collaboratively train a global model, fostering extensive knowledge exchange. However, this approach can lead to performance degradation in certain non-IID scenarios due to mutual interference during joint training.

Notably, FedLoRA significantly outperforms all baseline methods in the majority of scenarios, particularly as $\alpha$ increases. FedLoRA achieves this by effectively decoupling general and client-specific knowledge through parameter decomposition and mitigating the impact of non-IID through alternating training of full-rank and low-rank matrices.

## 4.3 ABLATION STUDIES

**Effect of $R_l$ and $R_c$.** As we discuss in Section 3.3, $R_l$ and $R_c$ individually denote the ratio of the low-rank matrix's rank to the full-rank matrix's rank in convolutional and fully-connected layers,

Table 2: The effect of $R_l$ and $R_c$ on CIFAR-10, CIFAR-100, and Tiny Imagenet under Dirichlet non-IID with $\alpha = 0.1$.

| Dataset | $R_c$ / $R_l$ | 20% | 40% | 60% | 80% | 100% |
|---|---|---|---|---|---|---|
| CIFAR-10 | 20% | $84.72 \pm 2.07$ | $84.97 \pm 1.74$ | $84.73 \pm 2.33$ | $84.80 \pm 2.03$ | $84.99 \pm 2.19$ |
| | 40% | $84.84 \pm 2.19$ | $84.96 \pm 1.87$ | $85.27 \pm 2.04$ | $84.97 \pm 1.86$ | $85.39 \pm 2.01$ |
| | 60% | $84.92 \pm 1.90$ | $85.35 \pm 1.96$ | $\mathbf{85.47 \pm 2.06}$ | $85.07 \pm 2.26$ | $85.38 \pm 1.76$ |
| | 80% | $84.70 \pm 1.98$ | $85.05 \pm 1.66$ | $85.25 \pm 2.00$ | $85.01 \pm 1.90$ | $85.13 \pm 1.95$ |
| | 100% | $85.09 \pm 1.95$ | $85.23 \pm 1.89$ | $85.15 \pm 1.66$ | $84.88 \pm 1.77$ | $85.21 \pm 1.62$ |
| CIFAR-100 | 20% | $62.00 \pm 0.60$ | $62.66 \pm 0.37$ | $61.99 \pm 0.97$ | $62.48 \pm 0.30$ | $62.70 \pm 0.83$ |
| | 40% | $61.70 \pm 0.28$ | $62.49 \pm 0.77$ | $62.70 \pm 0.59$ | $\mathbf{63.65 \pm 0.53}$ | $62.73 \pm 0.60$ |
| | 60% | $61.71 \pm 0.30$ | $62.88 \pm 0.33$ | $62.46 \pm 0.53$ | $63.12 \pm 0.38$ | $63.24 \pm 0.66$ |
| | 80% | $60.76 \pm 0.14$ | $62.54 \pm 0.56$ | $62.74 \pm 0.54$ | $62.15 \pm 0.38$ | $62.70 \pm 0.57$ |
| | 100% | $59.54 \pm 0.98$ | $60.97 \pm 0.35$ | $61.86 \pm 0.74$ | $61.96 \pm 0.55$ | $62.58 \pm 0.51$ |
| Tiny Imagenet | 20% | $40.77 \pm 0.10$ | $42.71 \pm 0.59$ | $43.27 \pm 0.43$ | $43.78 \pm 0.70$ | $43.88 \pm 0.16$ |
| | 40% | $40.14 \pm 0.36$ | $42.74 \pm 0.46$ | $43.82 \pm 0.46$ | $\mathbf{44.22 \pm 0.55}$ | $43.72 \pm 0.24$ |
| | 60% | $39.39 \pm 0.26$ | $42.75 \pm 0.46$ | $43.44 \pm 0.43$ | $43.85 \pm 0.90$ | $44.16 \pm 0.48$ |
| | 80% | $36.90 \pm 0.30$ | $41.94 \pm 0.29$ | $42.75 \pm 0.49$ | $43.21 \pm 0.41$ | $43.28 \pm 0.36$ |
| | 100% | $33.90 \pm 0.91$ | $40.55 \pm 0.15$ | $41.75 \pm 0.81$ | $42.16 \pm 0.41$ | $42.75 \pm 0.31$ |

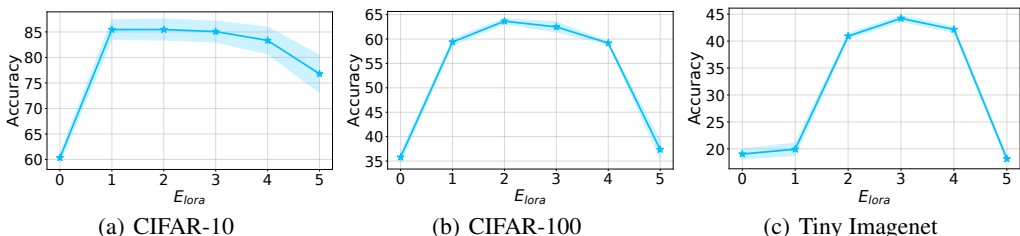

      (a) CIFAR-10             (b) CIFAR-100             (c) Tiny Imagenet

Figure 4: Effect of $E_{\text{lora}}$ in Dirichlet non-IID scenario with $\alpha = 0.1$.

respectively. They are two important hyper-parameters to control the learning ability of the low-rank matrices. In this section, we evaluate the effect of $R_l$ and $R_c$ on model accuracy. We choose $R_l$ and $R_c$ from $\{20\%, 40\%, 60\%, 80\%, 100\%\}$.

The experimental results are presented in Table 2. Firstly, we observe that the optimal combinations of $(R_c, R_l)$ are $(60\%, 60\%)$ for CIFAR-10, $(80\%, 40\%)$ for CIFAR-100, $(80\%, 40\%)$ for Tiny Imagenet. This underscores the importance of setting the personalized parameter matrices to low rank. Secondly, regarding the optimal combination as the focal point, model accuracy gradually decreases as the rank increases. This occurs because, after this point, the personalized matrices gain more learning capacity and begin to acquire some of the general knowledge. As a result, collaboration among clients on the shared matrices diminishes. As the rank decreases, model accuracy also gradually declines. This is because the personalized matrices fail to capture sufficient client-specific knowledge. This aligns with our expectations. Thirdly, experimental results highlight that model accuracy is more sensitive to changes in $R_l$ than $R_c$. This suggests that the acquisition of client-specific knowledge has a stronger correlation with the classifier than the feature extractor, consistent with prior research such as FedPer, FedRep, and FedRoD.

**Effect of $E_{\text{lora}}$ and $E_{\text{global}}$.** In this section, we verify the effect of $E_{\text{lora}}$ and $E_{\text{global}}$ on model accuracy. For simplicity, we set $E_{\text{global}} = E - E_{\text{lora}}$ and only adjust the value of $E_{\text{lora}}$. We conduct experiments on three datasets under Dirichlet non-IID with $\alpha = 0.1$ and sample $E_{\text{lora}} \in [0, E]$.

The experimental results are depicted in Fig. 4. When the $E_{\text{lora}} = 0$, FedLoRA essentially degenerates to FedAvg, and the accuracy closely resembles the FedAvg accuracy presented in Table 1, as expected. As $E_{\text{lora}}$ increases, the accuracy initially rises and then declines. When $E_{\text{lora}} = 5$, FedLoRA degenerates to local training with low-rank parameter matrices. However, due to the con-

Table 3: The effect of alternating training in FedLoRA on three datasets.

| METHODS | CIFAR-10 | CIFAR-100 | TINY |
|---|---|---|---|
| SIMULTANEOUSLY | $85.45 \pm 1.83$ | $61.18 \pm 1.05$ | $19.61 \pm 0.58$ |
| ALTERNATINGLY | $\mathbf{85.47 \pm 2.06}$ | $\mathbf{63.65 \pm 0.53}$ | $\mathbf{44.22 \pm 0.55}$ |

Table 4: The effect of LoRA matrices on model capacity.

| DATASETS & MODEL | LOCAL | LOCAL W/ LORA | FEDAVG | FEDAVG W/ LORA |
|---|---|---|---|---|
| CIFAR-10 & RESNET-8 | $81.91 \pm 3.09$ | $81.97 \pm 2.62$ | $60.39 \pm 1.46$ | $60.91 \pm 0.53$ |
| CIFAR-100 & RESNET-10 | $47.61 \pm 0.96$ | $47.64 \pm 0.79$ | $34.91 \pm 0.86$ | $35.91 \pm 0.70$ |

straints imposed by these low-rank matrices on the model's learning capacity, FedLoRA performs less effectively compared to the Local as shown in Table 1.

**Effect of Alternating training.** As we discussed in Section 3.4, different from previous work that trains personalized and shared components simultaneously, we propose to train the personalized part first and then the global part to reduce the impact of non-IID and better extract general knowledge. To evaluate this idea, in this experiment, we compare the performance of two training methods.

The experimental results on three datasets are shown in Table 3. We can see that when the learning task is simple (e.g., a 10-classification task on CIFAR-10), the performance of alternating training and simultaneous training of two matrices is similar. As the learning task becomes increasingly difficult, the performance improvement brought about by alternating training becomes more apparent. This is because, in the case of a simple learning task, the variations in tasks among clients are relatively minor, which facilitates the extraction of general knowledge. However, as the learning task complexity increases, the differences in tasks among clients gradually expand, rendering the extraction of general knowledge more susceptible to non-IID effects. In such scenarios, the utilization of our proposed alternating training method becomes increasingly crucial.

**Effect of Model Capacity.** In FedLoRA, we employ an additive decomposition technique on the model. In theory, this approach should not alter the model's capacity. However, in practical implementation, the decomposed model introduces low-rank matrices, thereby increasing the number of trainable parameters. This augmentation raises questions about whether the decomposed model genuinely enhances the model's capacity and whether the observed performance improvement is primarily a result of the increased number of trainable parameters. To address these concerns, we conducted an experiment to assess the impact on model capacity.

We conducted experiments using two configurations: CIFAR-10 with the ResNet-8 model and CIFAR-100 with the ResNet-10 model. We established two controlled scenarios: 1) 'Local' and 'Local w/ LoRA' indicate models without and with LoRA matrices that are exclusively trained locally. 2) 'FedAvg' and 'FedAvg w/ LoRA' indicate models without and with LoRA matrices trained using the FedAvg algorithm. The experimental results are shown in Table 4. Notably, we observe that, in comparison to the original model, the model enhanced with low-rank matrices exhibits only minimal performance improvement. This outcome underscores that our utilization of parameter decomposition does not bring about significant alterations to the model's capacity. Hence, the performance gains achieved by FedLoRA are not solely attributed to modifications in the model itself.

## 5 CONCLUSION

In this paper, drawing inspiration from the area of fine-tuning LLMs, we propose a new PFL method named FedLoRA. FedLoRA decomposes each model parameter matrix into a shared full-rank matrix and a personalized low-rank matrix. To further enhance the acquisition of general knowledge, we devise a training strategy that prioritizes the training of the low-rank matrix to absorb the influence of non-IID during local training. Our extensive experimental evaluations, conducted across multiple datasets characterized by varying degrees of non-IID, unequivocally demonstrate the superior performance of our FedLoRA method when compared to SOTA methods.

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

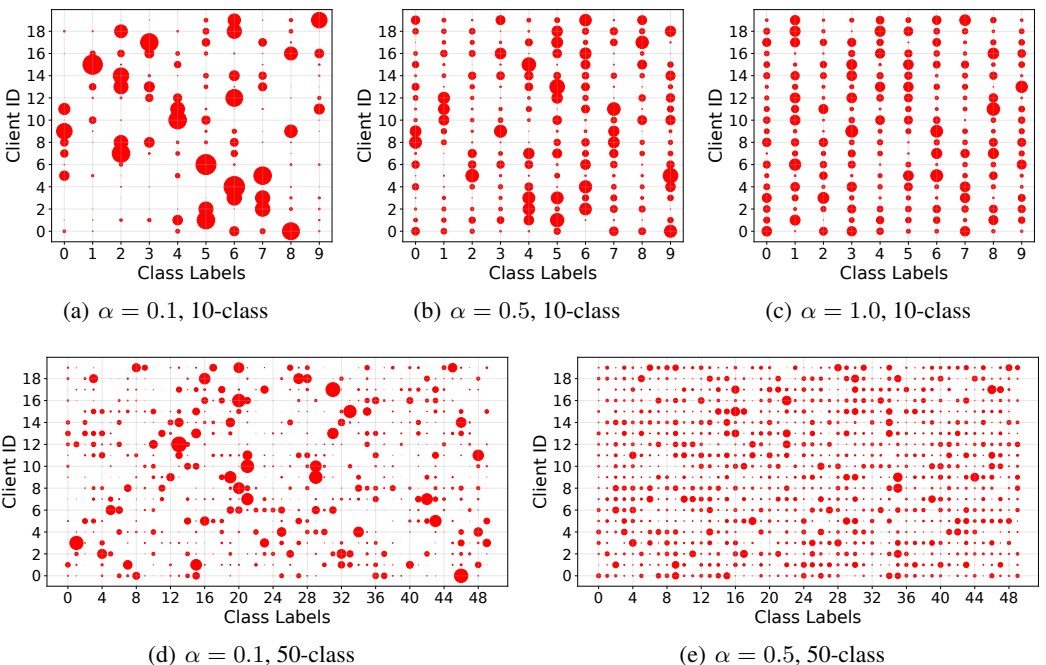

Figure 5: Visualization of data partitioning in Dirichlet non-IID scenarios with different $\alpha$.

Table 5: The effect of partial client participation.

| DATASETS | 100% | 90% | 70% | 50% |
|---|---|---|---|---|
| CIFAR-10 | 85.47±2.06 | 85.38±1.62 (-0.09) | 85.25±1.67 (-0.22) | 85.36±1.67 (-0.11) |
| CIFAR-100 | 63.65±0.53 | 63.01±0.10 (-0.64) | 63.21±0.18 (-0.44) | 63.13±1.05 (-0.52) |
| TINY | 44.22±0.55 | 44.13±0.70 (-0.09) | 44.10±0.26 (-0.12) | 43.99±0.62 (-0.23) |

## A  VISUALIZATION OF DATA PARTITIONING IN DIRICHLET NON-IID

To facilitate intuitive understanding, we utilize 20 clients on the 10-classification and 50-classification datasets to visualize the data distribution of clients with different $\alpha$ values. As shown in Figure 5, the horizontal axis represents the data class label index, and the vertical axis represents the client ID. Red dots represent the data assigned to clients. The larger the dot is, the more data the client has in this class. When $\alpha$ is small (e.g., $\alpha = 0.1$), the overall data distributions of clients vary greatly. However, the variety of client data distribution is low, and it is easy to have clients with very similar data distributions. As the $\alpha$ increases, the extent of class imbalance within each client's dataset gradually diminishes, consequently leading to more difficult local tasks (i.e., the number of classes involved and a reduction in the number of samples available for each class). Concurrently, the dissimilarity in data distribution among different clients gradually diminishes, while the diversity in client data distribution widens. Furthermore, comparing the 10-classification dataset and the 50-classification dataset, it can be seen that under the same $\alpha$ value, when the number of dataset classes increases, the difference of client data distribution becomes larger, and the diversity of client data distribution increases. It becomes more difficult to extract general knowledge among clients.

In summary, the Dirichlet non-IID configuration proves to be a potent approach for assessing the performance of PFL methods across a spectrum of intricate and diverse non-IID scenarios.

## B  PARTIAL CLIENT PARTICIPATION

In our previous experiments, we assume all clients participate in FL training in each round. However, some clients may be offline due to reasons such as unstable communication links. This is the partial

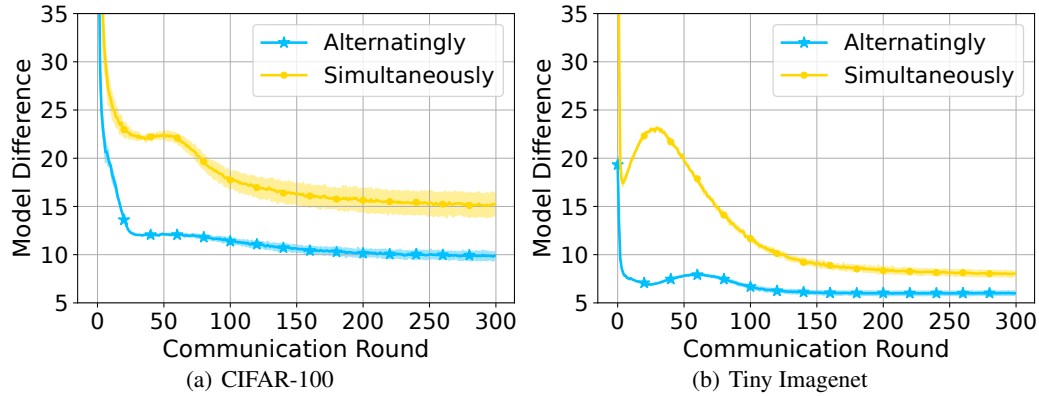

Figure 6: Effect of training logic on average model difference of $\sigma_i, 1 \le i \le N$ and $\overline{\sigma}$ in Dirichlet non-IID scenario with $\alpha = 0.1$.

client participation problem that is common in FL. In this section, we evaluate the robustness of FedLoRA to this problem. We consider the scenarios where 90%, 70%, and 50% clients participate in each round and carry experiments on CIFAR-10, CIFAR-100, and Tiny Imagenet with $\alpha = 0.1$.

The results are illustrated in Table 5. As we can see, in all scenarios, partial client participation does not significantly affect accuracy compared to all client participation. This is attributed to FedLoRA's effectively separating general knowledge and client-specific knowledge, and the effect of non-IID is reduced through alternate training. Ensure that client collaboration is not significantly affected by outline clients in each round.

## C    THE EFFECT OF ALTERNATING TRAINING ON MODEL DIFFERENCE

As we discussed in Section 3.4 and Fig. 3, the primary objective of alternating training is to mitigate the impact of data heterogeneity on the shared parameters, essentially reducing the deviation of shared parameters to the local minimum point of the client. Consequently, employing alternating training should lead to a reduction in the discrepancies among shared parameters across clients during their local training phases. To validate the effectiveness of alternating training in achieving this goal, we carried out additional experiments to compare the disparities in shared parameters among clients when using alternating training as opposed to not using it. Specifically, we calculate the average model distance between $\sigma_i, 1 \le i \le N$ and $\overline{\sigma}$ by

$$\frac{1}{N} \sum_i^N \|\sigma_i^t - \overline{\sigma}^t\|_2 \tag{9}$$

in each round $t$. The results are shown in Fig. 6. It is evident from the data that, across both datasets, the utilization of alternating training significantly diminishes the differences in the shared parameters among clients. This is consistent with our analysis in the paper.

## D    WHETHER FEDLORA SACRIFICES SOME CLIENTS' ACCURACY

In previous experiments, we demonstrate the improvement of the averaged accuracy of all clients. In this section, we focus on the individual improvement for each client and verify whether FedLoRA sacrifices some clients' accuracy. We plot each client's accuracy in FedLoRA, FedAvg, and Local (i.e., each client trains the model locally without collaboration) methods in the Dirichlet non-IID scenario with $\alpha = 0.1$. The results are shown in the Fig. 7.

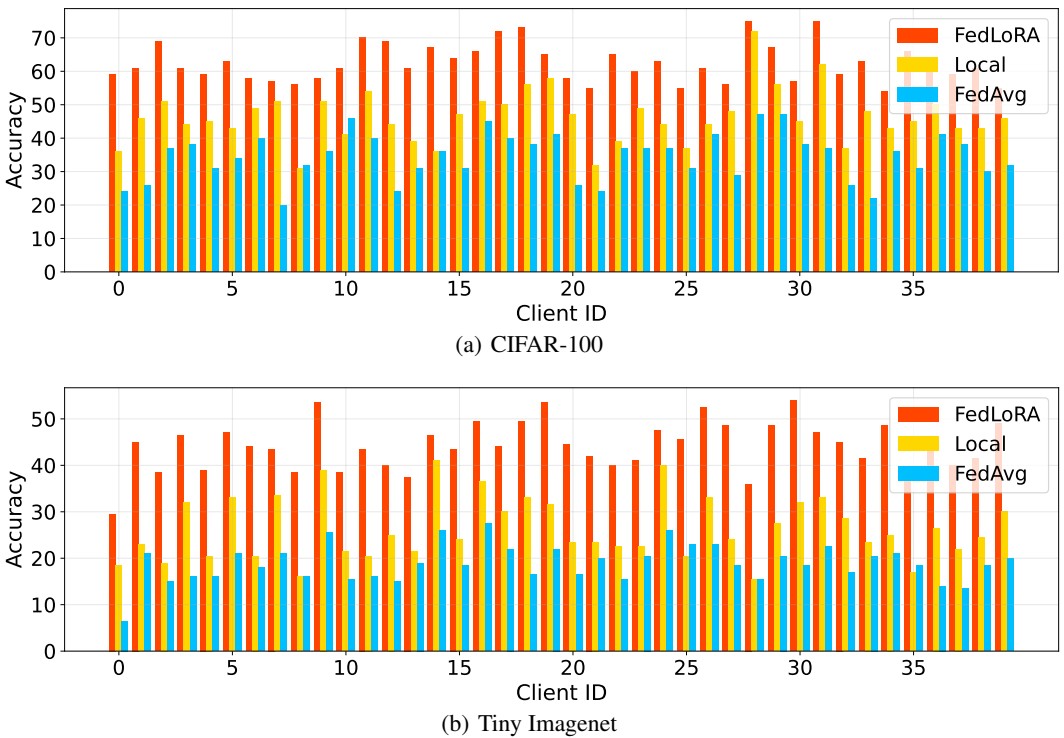

Figure 7: Test accuracy of each client in Dirichlet non-IID scenario with $\alpha = 0.1$.

Table 6: Comparison results on larger datasets.

| METHODS | FEDAVG | FEDPER | FEDROD | FEDLORA |
|---|---|---|---|---|
| AG_NEWS | 89.36 | 90.76 | 91.38 | **91.79** |
| IMAGENET-SUBSET | 18.55 | 29.37 | 32.45 | **35.67** |

Notice that the accuracy of all clients in the FedLoRA method is higher than that in the FedAvg and Local methods, affirming that the use of FedLoRA does not lead to any deterioration in individual client performance.

## E   ADDITIONAL EXPERIMENTS ON LARGER DATASETS

While the current mainstream FL work focuses on the algorithm's performance on small image datasets, in this section, we further verify the performance of FedLoRA on larger datasets as well as other modality datasets.

Specifically, we conduct additional experiments on both a larger image dataset and a natural language processing (NLP) dataset. For the larger image dataset, we selected a subset from ImageNet, consisting of 400 classes with a total of 80,000 samples. We utilized the ResNet-10 model architecture, with each client having 2,000 training samples generated following the Dirichlet distribution with $\alpha = 0.1$. For the NLP dataset, we opted for AG_NEWS, a text 5-classification dataset with 120,000 samples. We employed the Transformer model architecture, with each client having 3,000 training samples generated following the Dirichlet distribution with $\alpha = 1.0$. Additionally, for the Transformer model, we applied model decomposition to the weights in the self-attention modules and fully connected weights in the classifier module.

Table 6 displays the test accuracy results for these two datasets. It's evident that FedLoRA consistently outperforms other state-of-the-art methods on both datasets.