# OpenReview forum: "FedLoRA: When Personalized Federated Learning Meets Low-Rank Adaptation"
_ICLR.cc/2024/Conference — Submitted to ICLR 2024_

### Official Review · Reviewer_yWWG · 2023-11-01

**Soundness:** 3 good
**Presentation:** 3 good
**Contribution:** 3 good
**Rating:** 6
**Confidence:** 3

**Summary:**

This paper provide a personalized federated learning FedLoRA. The main idea behind this approach is to decompose each client's personalized model into two components: a full-rank matrix and a low-rank matrix capturing client-specific knowledge. It is a useful method in specific scenario.

**Strengths:**

FedLoRA emphasizes personalizing models for each client, which helps in catering to the unique data distribution of each client.
FedLoRA efficiently decomposes model parameters into full-rank (general) and low-rank (client-specific) matrices. And it uses an alternating training strategy to train low-rank and full-rank parameters.

**Weaknesses:**

Even though the low-rank parameters remain private, sharing the full-rank parameters with the server might raise privacy concerns for some applications.
Compared to LoRA, the complexity and memory cost might be high.

**Questions:**

Could the author provide details on the memory cost of this method compared to the general LoRA method?

---

> ### Author Response · Authors · 2023-11-19
>
> ## Q1: ... sharing the full-rank parameters with the server might raise privacy concerns ...
>
> Thank you for your feedback. It seems there might have been some misunderstanding. In fact, the core principle of federated learning is to ensure privacy preservation by sharing full-rank model parameters. Our proposed FedLoRA adheres to this same design, sharing full-rank model parameters with the server, and as such, it does not compromise privacy preservation.
>
> With that said, it's important to note that some research works do recognize the potential privacy leakage risk associated with parameter transmission and propose specific solutions to enhance privacy preservation. These methods can be seamlessly integrated into our existing FedLoRA approach. However, since our study does not primarily focus on this aspect, we have not undertaken additional designs and experiments in this regard.
>
> ## Q2: ... provide details on the memory cost ... compared to the general LoRA ...
>
> Thanks for the comment. While we draw inspiration from the design principles of LoRA, it's essential to note that the application scenarios, actual methods, and objectives of our approach are significantly different from those of LoRA. **Hence, we believe that making direct comparisons between FedLoRA and LoRA might not be valid.**. Specifically, LoRA aims to train a low-rank part for fine-tuning a given pre-trained model (full rank). In contrast, FedLoRA aims to train a full-rank model for extracting shared knowledge and simultaneously train a low-rank personalized part for each client.
>
> We believe that it would be more appropriate to compare FedLoRA, specifically in terms of computational cost, with other federated learning algorithms, such as FedAvg. Benefiting from our proposed alternating training, FedLoRA has a **lower computation overhead** than most traditional FL methods such as FedAvg. For example, if all clients in FL need to train locally for $E$ epochs before uploading parameters, then for FedAvg, each epoch requires training a standard full-rank model; but for FedLoRA, it trains $E_{lora}$ rounds of the low-rank part (fewer trainable parameters) and $E - E_{lora}$ rounds of the full-rank part.  Therefore, the overall computational overhead is actually reduced compared to FedAvg.

---

> ### Author Response · Authors · 2023-11-23
>
> Dear respected reviewer, we are wondering whether you have any further questions regarding our work? We posted our response regarding your concerns on 19st Nov but have not received your valuable comments yet.  Any responses or score updates will be greatly appreciated.

---

> > ### Comment · Reviewer_yWWG · 2023-11-23
> >
> > Thank you for the author's response; I would like to maintain my score.

---

### Official Review · Reviewer_fKpJ · 2023-11-03

**Soundness:** 2 fair
**Presentation:** 3 good
**Contribution:** 2 fair
**Rating:** 3
**Confidence:** 5

**Summary:**

The paper addresses data heterogeneity within Personalized Federated Learning (PFL) by leveraging low-rank approximation. Specifically, each client retains a low-rank matrix locally to encapsulate personalized information derived from private data. In contrast, a full-rank matrix is updated and aggregated to capture more generalized information. The effectiveness of this approach is demonstrated through extensive experimentation.

**Strengths:**

1. Utilizing low-rank approximation for storing personalized information is an innovative approach to managing data heterogeneity in PFL.

2.The paper introduces an alternative training strategy to improve the overall aggregated performance, which could be a meaningful contribution to the field.

3.Different prototypes for the decomposition of CNN and dense layers are presented, broadening the scope of the proposed method.

4.The extensive experimental results show the effectiveness of the proposed method.

**Weaknesses:**

1.The paper could benefit from a more profound exposition of the underlying intuition of the proposed method. Offering analytical comparisons or insights on why this method outperforms state-of-the-art approaches could enrich the narrative.

2.The concern of overfitting associated with the training of the full-rank matrix needs to be adequately addressed. This aspect could affect the generalization capability of the aggregated parameters.

3.The method's scalability seems constrained by model size. As model size increases, transferring the full-rank matrix to the server becomes impractical.

**Questions:**

Weaknesses

---

> ### Author Response · Authors · 2023-11-19
>
> ## Q1: ... benefit from more profound exposition of the underlying intuition of the proposed method. Offering analytical comparisons or insights why this method outperforms state-of-the-art approaches  ...
> Thanks for your review.
>
> **The insight of applying LoRA to PFL:**
>
> Benefiting from the low-rank nature of LoRA, introducing LoRA into FL for personalization has the following advantages over current PFL methods. First, constraining the personalized part to low-rank can **reduce the overfitting to the local knowledge**, especially when the clients' local data are very limited.  Second, the low-rank structure allows the personalized part to **focus its learning on the most critical aspects of the local knowledge.** Third, LoRA greatly reduces the trainable parameters, it helps in **retaining much of the general knowledge** acquired from other clients to enhance generalization.
>
> **The insight of proposed alternating training:**
>
> The motivation for proposing alternating training is that by first training the personalized part $\tau$, the client-specific knowledge is mostly learned by $\tau$ and the shift of $\theta$ to the local minimum point is mainly done by $\tau$. Therefore, **alternating training can reduce the deviation of shared parameters $\sigma$ to the local minimum point of the client,** thereby reducing the influence of non-IID on general knowledge extraction. This motivation has been discussed in Section 3.4.
>
> To validate this, we add an experiment to further prove this intuition. We add additional experiments to compare the differences in shared parameters among clients with and without alternating training. We expect that if alternating can reduce the deviation of $\sigma$, then the difference of the shared part among clients in the local training phase should be reduced.  The results are shown in the table below. We can see that on both datasets, using alternating training can significantly reduce the difference in the shared part of clients.
>
> |                             | CIFAR-100 | Tiny Imagenet |
> | --------------------------- | --------- | ------------- |
> | Simultaneous Training       | 19.05     | 12.25         |
> | Alternating Training (ours) | 11.74     | 6.59          |
>
> For more detailed statistics including the trend of model difference during the training process, you can refer to Appendix C in our revised version.
>
> We intend to incorporate these experiments into a revised version of the paper, and we are committed to making all the source code publicly available in the future.
>
> ## Q2: The concern of overfitting ... of the full-rank matrix needs to be adequately addressed ...
> Thanks for your review.
>
> We feel that there might be some miscommunication here. In fact,  **FedLoRA has the same amount of shared parameters compared with the baseline method FedAvg**. Therefore, FedLoRA does not bring additional risks of over-fitting. Here are detailed explanations:
>
> First, we would like to clarify that our intention is not to use LoRA for training large language models (LLMs) in Federated Learning (FL). Instead, we draw inspiration from the concept of LoRA, which was originally conceived for fine-tuning LLMs, and adapt it to the context of Personalized Federated Learning (PFL). Consequently, the full-rank model employed in FedLoRA essentially resembles a traditional neural network commonly used in the FL literature. Therefore, FedLoRA has **the same amount of shared parameters** compared to the majority of FL methods that upload models, such as FedAvg. As a result, our FedLoRA does not bring additional risks of over-fitting.
>
> ## Q3: The method's scalability seems constrained by model size ...
> Thanks for your review.
>
> This question has been partially addressed in our response to Q2. In our reply to Q2, we emphasize that **FedLoRA maintains an equivalent number of shared parameters compared to the baseline method**, FedAvg. Consequently, in comparison to the baseline algorithm, FedAvg, our approach does not incur additional communication costs when applied to a large-sized model.

---

> ### Author Response · Authors · 2023-11-23
>
> Dear respected reviewer, we are wondering whether you have any further questions regarding our work? We posted our response regarding your concerns on 19st Nov but have not received your valuable comments yet.  Any responses or score updates will be greatly appreciated.

---

> > ### Comment · Reviewer_fKpJ · 2023-11-23
> > **Thanks for the responses**
> >
> > Thank you for the rebuttal. My main concerns are still unaddressed, such as the strength and scalability of the method. Iterative back-and-forth full-rank matrix update and transmission raise a practical efficiency issue, compared with existing parameter-efficient fine-tuning algorithms, which adjust only a small portion of parameters. In addition, there are no efficiency tests found. I would like to keep my rating.

---

> ### Author Response · Authors · 2023-11-23
>
> We sincerely appreciate the comments provided by the reviewer. To clarify, we need to reiterate that **our goal is not to fine-tune a given pre-trained network**; rather, it is to **train a model from scratch in the context of federated learning**. This task is the most common task setting in the field of federated learning, and the model trained from scratch is full-rank in the vast majority of scenarios. Consequently, the appropriate benchmark for our work should not be the parameter-efficient fine-tuning of models. Rather, it should involve the comparison with similar efforts aimed at training full-rank models from scratch, which in the field of federated learning, refers to FedAvg. Additionally,  in response to the reviewer's pertinent query about the computational costs involved, we list detailed numerical results in our ResNet-10 experiments here for reference:
>
> | ResNet-10                       | FedAvg                | FedLoRA                                                   |
> | ------------------------------- | --------------------- | --------------------------------------------------------- |
> | Number of parameters to update | $E \times 11.24M$ | $E_{lora} \times 8.58M +(E - E_{lora}) \times 11.24M$ |
> | Number of parameters to upload | $11.24M$ | $11.24M$ |
>
> In the above table, 'Number of parameters to update' refers to the total number of updates in each training round (i.e., the number of updated parameters $\times$ update times), which directly reflects the computational cost in each client.
>
> In our experiments, the number of full-rank parameters (shared) is 11.24M, and the number of low-rank parameters (personalized) is 8.58M. The $E=5$ and the $E_{lora} = 3$. Hence the final number of parameters to update for FedAvg is 56.2M, and that for FedLoRA is 48.22M. Therefore, compared with the FedAvg algorithm, **our computation overhead is lower and the communication overhead is the same.**

---

### Official Review · Reviewer_8LH2 · 2023-11-08

**Soundness:** 3 good
**Presentation:** 4 excellent
**Contribution:** 4 excellent
**Rating:** 8
**Confidence:** 3

**Summary:**

This paper introduces FedLoRA that uses LoRA for personalized federated learning (PFL). The main idea is that we decompose the full-rank model $\theta$ to a full-rank shared parameters $\sigma$ and low-rank personalized parameters $\tau$ as $\theta = \sigma + \tau$. On each communication round, the clients would first freeze $\sigma$ and optimize $\tau$, then freeze $\tau$ and optimize $\sigma$ (alternative optimization). Then clients would send $\sigma$ to the server to average $\sigma$ on the server side. The server would broadcast the averaged $\bar{\sigma}$ back to each client.

The experiments are ResNet on CIFAR10, CIFAR-100, and TinyImageNet. The authors consider the Dirichlet non-IID client data distribution with $\alpha \in \{0.1, 0.5, 1\}$ and demonstrate FedLoRA's superiority over SOTA PFL methods. The authors also ablate on (1) the rank of convolution or linear layers for $\tau$ (2) training epochs for $\sigma$ and $\tau$ per communication round (3) alternative optimization vs. simultaneous optimization (4) performance boost from introducing $\tau$.

**Strengths:**

The motivation behind using LoRA for simultaneously learning client local knowledge and mitigate the data heterogenity issue for FL as we use the additional full-rank weights for averaging is quite good.

Most of the ablation studies are well-formulated and well-executed.

Most of the paper is clearly written and easy to follow.

**Weaknesses:**

I cannot find any major weakness of this paper, but there is a claim without an support from strong evidence:
- In the paragraph of 'Effect of $R_l$ and $R_c$' of section 4.3, the authors claim that the decrease of model accuracy w.r.t. the rank of $\tau$ after the focal point is because $\tau$ acquires some of the general knowledge. To support this argument, we should compute the vector similarity of $\tau$ across each client and we should see an increase after the focal point.

**Questions:**

Is it possible to extend the experiments to use a ImageNet-pretrained ResNet to finetune on CIFAR10/CIFAR100? I assume in this paper we need to learn $\sigma$ because we are training from scratch and we still need to learn the feature extractor. I expect that if the feature extractor is already trained (as a ImageNet-pretrained model), we should expect that $\tau$ is learned more often than $\sigma$. ($\Delta \sigma$ should be much smaller than $\Delta \tau$). This study could be a strong support on the client-specific knowledge statement.

This is an overall good paper and I will vote for accept.

---

> ### Author Response · Authors · 2023-11-19
>
> ## Q1: Is it possible to extend the experiments to use a ImageNet-pretrained ResNet ... we should expect that $\tau$ is learned more often than $\sigma$ ...
>
> Thanks for your review. As requested by the reviewer, we add experiments about how the update for global part and personalized part (denoted as $\Delta \sigma$ and $\Delta \tau$) affect the final performance. Specifically, we initialize the $\sigma$ with ImageNet pre-trained weights and conduct experiment on the CIFAR-100 dataset in the Dirichlet non-IID scenario with $\alpha=0.1$.
>
> We control the value of $E_{lora}$ (higher $E_{lora}$ means update $\tau$ more often, and results in larger $\Delta \tau$) at different levels. The results are summarized in the table below.
>
> | $E_{lora}$      | 0     | 1     | 2     | 3     | 4     |
> | --------------- | ----- | ----- | ----- | ----- | ----- |
> | Accuracy        | 39.35 | 50.37 | 71.47 | 72.00 | 72.42 |
> | $\Delta \sigma$ | 37.99 | 31.59 | 22.12 | 16.16 | 10.57 |
> | $\Delta \tau$   | 0.00  | 26.48 | 89.68 | 98.26 | 99.63 |
>
> From this table, we can conclude that **when $\Delta \tau$ is larger than $\Delta \sigma$, FedLoRA achieves better results**. This indicates that with the pre-trained weights, the $\tau$ should be updated more often than $\sigma$ to learn client-specific knowledge. This aligns with the reviewer's expectation and can serve as strong support for the client-specific knowledge statement.
>
> We intend to incorporate these experiments into a revised version of the paper, and we are committed to making all the source code publicly available in the future.

---

> ### Author Response · Authors · 2023-11-23
>
> Dear respected reviewer, we are wondering whether you have any further questions regarding our work? We posted our response regarding your concerns on 19st Nov but have not received your valuable comments yet.  Any responses or score updates will be greatly appreciated.

---

> ### Comment · Reviewer_8LH2 · 2023-11-23
> **Thank you for this additional experiment**
>
> **Q1**
>
> This ImageNet-pretrained ResNet experiment on CIFAR100 looks good! I don't have any other major concerns about this paper now.
>
> I would like to maintain my score.

---

### Official Review · Reviewer_eXdF · 2023-11-08

**Soundness:** 2 fair
**Presentation:** 3 good
**Contribution:** 3 good
**Rating:** 5
**Confidence:** 5

**Summary:**

This paper proposed FedLoRA which decomposes shared and personalized parameters like LoRA in fine-tung LLMs, and employ a new training strategy to optimize it in non-IID settings.

**Strengths:**

This paper innovatively introduced LoRA as a personalized model and a new training strategy to mitigate the data heterogeneity problem for general knowledge learning.

This paper is written in a well-organized way for easy understanding and following.

**Weaknesses:**

The paper lacks experiments and analysis about training and communication cost, since the new training strategy needs to train two times for full-rank matrix and low-rank matrix, and the cost for transmitting full-rank matrixes may be huge because they are much larger than low-rank matrix.

This paper does not include the evaluation results of each client which may lead to sacrificing the performance of some clients for improvement.

The convergence analysis is missing in this paper, and there are no experiments and analysis about the proposed new training stage to support the intuition in Fig 3.

**Questions:**

Please refer to the weakness.

---

> ### Author Response · Authors · 2023-11-19
>
> ## Q1: ...lacks experiments and analysis about training and communication cost ... the new training strategy needs to train two times ... the cost for transmitting full-rank matrixes may be huge ...
>
> Thank you for your feedback. We feel that there might be some miscommunication here. We would like to emphasize that FedLoRA has **the same communication overhead** and **lower computation overhead** than the baseline method FedAvg. Here are the detailed explanations:
>
> First, we would like to clarify that our intention is not to use LoRA for training large language models (LLMs) in Federated Learning (FL). Instead, we draw inspiration from the concept of LoRA, which was originally conceived for fine-tuning LLMs, and adapt it to the context of Personalized Federated Learning (PFL). Consequently, the full-rank model employed in FedLoRA essentially resembles a traditional neural network commonly used in the FL literature. As a result, FedLoRA shares the same communication overhead as the majority of FL methods since these methods, including FedAvg, all involve uploading the full-rank model.
>
> In terms of computational cost, it's worth noting that FedLoRA actually incurs a lower computational overhead compared to many traditional FL methods, including FedAvg. This advantage can be attributed to our proposed alternating training technique. For example, in a FL setting where all clients need to train locally for $E$ epochs before uploading parameters, then for FedAvg, each epoch requires training a standard full-rank model; but for FedLoRA, it trains $E_{lora}$ rounds of the low-rank part (fewer trainable parameters) and $E - E_{lora}$ rounds of the full-rank part.  Therefore, the overall computational overhead is actually reduced compared to FedAvg.
>
> ## Q2: ... may lead to sacrificing the performance of some clients for improvement.
>
> Thank you for your valuable insights. We have performed additional experiments to demonstrate that our proposed FedLoRA, which incorporates personalized low-rank parameter matrices, does not result in any degradation of individual client performance.
>
> The experimental details are as follows. We test FedLoRA on CIFAR-100 and Tiny-ImageNet datasets in the Dirichlet non-IID scenario with $\alpha=0.1$. Specifically, we plot the mean / minimum / maximum of accuracy improvements of FedLoRA for each client, compared with FedAvg and Individual Training (i.e., each client trains the model locally without collaboration).
>
> The results are summarized in the following table.
>
> | CIFAR-100 | Mean Value of Improvement | Minimum Value of Improvement | Maximum Value of Improvement |
> |-| - | - | - |
> | Compared with FedAvg | 27.97 | 15.00| 45.00|
> | Compared with Individual Training | 16.07| 3.00| 31.00|
>
> | Tiny Imagenet | Mean Value of Improvement | Minimum Value of Improvement | Maximum Value of Improvement |
> | - | - | - | - |
> | Compared with FedAvg | 25.29 | 18.50 | 35.50 |
> | Compared with Individual Training | 17.86 | 5.50 | 25.00 |
>
> For more detailed statistics for each client, please refer to Appendix D in our revised paper.
>
> Notice that the minimum improvement value across all clients when employing FedLoRA is greater than 0, affirming that the use of FedLoRA does not lead to any deterioration in individual client performance.
>
> We intend to incorporate these experiments into a revised version of the paper, and we are committed to making all the source code publicly available in the future.
>
> ## Q3: ... no experiments and analysis ... to support the intuition in Fig 3.
>
> Thanks for the comment. We conduct additional experiments to further validate the intuition in Fig 3. Here is the detailed experiment information.
>
> The motivation for proposing alternating training is to reduce the impact of data heterogeneity on the shared part. That is, **reduces the deviation of shared parameters to the local minimum point of the client**. Thus, we expect that using alternating training should reduce the difference in shared parameters among clients during local training. We conduct experiments to analyze the change in model distance before and after using alternating training. The results are shown in the following table:
>
> |                             | CIFAR-100 | Tiny Imagenet |
> | --------------------------- | --------- | ------------- |
> | Simultaneous Training       | 19.05     | 12.25         |
> | Alternating Training (ours) | 11.74     | 6.59          |
>
> For more detailed statistics including the trend of model difference during the training process, please refer to Appendix C in our revised paper. It is evident that the utilization of the alternating training strategy results in a significantly smaller disparity in the shared components of the model. This observation aligns with our intuition, as illustrated in Fig. (3).
>
> We intend to incorporate these experiments into a revised version of the paper, and we are committed to making all the source code publicly available in the future.

---

> ### Author Response · Authors · 2023-11-23
>
> Dear respected reviewer, we are wondering whether you have any further questions regarding our work? We posted our response regarding your concerns on 19st Nov but have not received your valuable comments yet.  Any responses or score updates will be greatly appreciated.

---

### Official Review · Reviewer_PiFj · 2023-11-10

**Soundness:** 2 fair
**Presentation:** 2 fair
**Contribution:** 2 fair
**Rating:** 5
**Confidence:** 3

**Summary:**

This paper applies LoRA to personalized federated learning and proposes an alternative training method for local clients. Experiments on three image datasets show the effectiveness of the proposed method.

**Strengths:**

1. Personalized federated learning, although extensively studied, is interesting.

2. The proposed method seems to achieve good results reported by this paper.

3. This paper is easy to read.

**Weaknesses:**

1. I cannot see too much novelty in applying LoRA to federated learning. It is not clear why LoRA is a better choice for personalized FL than existing methods. Moreover, several existing works have explored this, for example, FedLoRA: Model-Heterogeneous Personalized Federated Learning with LoRA Tuning, which also has the name of FedLoRA.

2. The proposed alternative training seems heuristic and lacks theoretical justification.

3. Experiments are conducted on three small image datasets. More large datasets, especially in different modalities,  should be used.

4. The paper writing quality should be improved. There are many issues, like "Federated learning (FL) McMahan et al. (2016) allows clients to collaboratively train a global model"

**Questions:**

Why not try larger datasets, and datasets beyond image?

---

> ### Author Response · Authors · 2023-11-19
>
> Thank you for your feedback.
>
> ## Q1: ... novelty in applying LoRA to federated learning. ... why LoRA is a better choice for personalized FL than existing methods ...
>
> First, recall that LoRA was initially designed for fine-tuning large language models (LLMs). Its core concept involves the integration of low-rank parameter matrices, which have fewer trainable parameters, into the pre-trained model to adapt it for downstream tasks. In a similar vein, in the context of personalized federated learning (PFL), FedLoRA plays a role by incorporating personalized low-rank matrices into the shared global model for each client. This enables the global model to acquire client-specific knowledge.
>
> Benefiting from the low-rank nature of LoRA, introducing LoRA into FL for personalization has the following advantages over current PFL methods. First, constraining the personalized part to low-rank can **reduce the overfitting to the local knowledge**, especially when the clients' local data are very limited.  Second, the low-rank structure allows the personalized part to **focus its learning on the most critical aspects of the local knowledge.** Third, LoRA greatly reduces the trainable parameters, it helps in **retaining much of the general knowledge** acquired from other clients to enhance generalization.
> ## Q2: several existing works have explored this ... has the name of FedLoRA
>
> The paper titled 'Model-Heterogeneous Personalized Federated Learning with LoRA Tuning' was made available on arXiv on October 20, which is after the submission date of our paper (e.g., September 28). Nevertheless, it's important to note that our paper differs significantly from this work in terms of objectives and methodologies. While they employ LoRA to address the challenge of model heterogeneity, our research focuses on addressing data heterogeneity. Their primary goal is to reduce computing and communication overhead, whereas our aim is to improve the separation between the learning of general knowledge and client-specific knowledge.
>
> ## Q3: ... alternative training ... lacks theoretical justification.
>
> While we reserve the theoretical justification for future work, we have taken your comment into consideration and conducted empirical experiments to validate the effectiveness of the proposed alternating training strategy.
>
> As a quick reminder, the primary objective of alternating training is to mitigate the impact of data heterogeneity on the shared parameters, essentially **reducing the deviation of shared parameters to the local minimum point of the client**. Consequently, employing alternating training should lead to a reduction in the discrepancies among shared parameters across clients during their local training phases. To validate the effectiveness of alternating training in achieving this goal, we carried out additional experiments to compare the disparities in shared parameters among clients when using alternating training as opposed to not using it. The results are presented in the table below. It's evident from the data that, across both datasets, the utilization of alternating training significantly diminishes the differences in the shared parameters among clients.
>
> || CIFAR-100 | Tiny Imagenet |
> | - | - | - |
> | Simultaneous Training| 19.05| 12.25|
> | Alternating Training (ours) |11.74| 6.59|
>
> For a more comprehensive set of statistics, including the trend of model differences throughout the training process, please refer to Appendix C in our revised paper.
>
> ## Q4: More large datasets, especially in different modalities, should be used.
>
> In response to your comment, we have conducted additional experiments on both a larger image dataset and a natural language processing (NLP) dataset.
>
> - For the larger image dataset, we selected a subset from ImageNet, consisting of 400 classes with a total of 80,000 samples. We utilized the ResNet-10 model architecture, with each client having 2,000 training samples generated following the Dirichlet distribution with $\alpha=0.1$.
> - For the NLP dataset, we opted for AG_NEWS, a text classification dataset with 120,000 samples. We employed the Transformer model architecture, with each client having 3,000 training samples generated following the Dirichlet distribution with $\alpha=1.0$. Additionally, for the Transformer model, we applied model decomposition to the weights in the self-attention modules and fully connected weights in the classifier module.
>
> | | FedAvg | FedPer | FedRoD | FedLoRA   |
> | -| - | - | - | - |
> | AG_NEWS | 89.36  | 90.76  | 91.38  | **91.79** |
> | ImageNet (subset) | 18.55  | 29.37  | 32.45  | **35.67** |
>
> The table above displays the test accuracy results for these two datasets. It's evident that FedLoRA consistently outperforms other state-of-the-art methods on both datasets. We intend to incorporate these experiments on the new datasets into a revised version of the paper, and we are committed to making all the source code publicly available in the future.

---

> ### Author Response · Authors · 2023-11-23
>
> Dear respected reviewer, we are wondering whether you have any further questions regarding our work? We posted our response regarding your concerns on 19st Nov but have not received your valuable comments yet.  Any responses or score updates will be greatly appreciated.

---

### Author Response · Authors · 2023-11-19

**Dear reviewers and area chair:**

We thank all the reviewers for their kind suggestions and acknowledgments of our work. Their suggestions are very important to enhance our work. For their rebuttal requests, we try our best to respond. We intend to incorporate all experiments conducted in the rebuttal phase into a revised version of the paper, and we are committed to making all the source code publicly available in the future.

---

### Meta-Review · Area_Chair_yPUE · 2023-12-16

**Metareview:**

The paper presents FedLoRA that uses LoRA for personalized federated learning (PFL). The main idea is to represent the model parameters as a summation of full-rank shared parameters and and low-rank personalized parameter. The shared and personal parameters are updated in an alternating manner and clients communicate shared parameters/gradients to the server and receive the averaged update. The experiments use ResNet on CIFAR10, CIFAR-100, and TinyImageNet with Dirichlet non-IID client data distribution. Results demonstrate FedLoRA superiority over other PFL methods in some cases.

Strengths:
+ The paper is well written and easy to understand
+ The idea of using LoRA for simultaneously learning client personalized knowledge and mitigate the data heterogenity issue for FL is interesting.

Weaknesses:
- Novelty of the method is limited as the method is a relatively simple application of LoRA
- Experiments are on small image datasets and the performance improvement is small.
- The choice of non-IID data from the same distribution seems a strange choice for PFL. I think an experiment with some experiments with strict separation between private and shared data can strengthen the paper.
- The paper lacks experiments and analysis about training and communication cost. While authors think this point is not important, reviewers and I think this is an important consideration for PFL.
- Privacy concerns are key part of federated learning (otherwise, it is just distributed learning that does not offer any privacy constraints). While I agree that privacy-preserving techniques can be integrated with FedLoRA, it would have been good to see some evidence and effects of those approaches.

**Justification For Why Not Higher Score:**

- Additional experiments and discussion on communication cost associated with parameter sharing, privacy issues for PFL, and datasets with clear separation between shared and private data would have strengthened the paper

**Justification For Why Not Lower Score:**

N/A

---

### Decision · Program_Chairs · 2024-01-16

Reject